# A new simple brain segmentation method for extracerebral intracranial tumors

**Xiaolin Hou⊙, Dongdong Yang⊙\*, Dingjun Li, Meijun Liu, Yuan Zhou, Min Shi**

School of Clinical Medicine, Hospital of Chengdu University of Traditional Chinese Medicine, Chengdu University of TCM, Chengdu, China

\* hxlhxf@126.com

**Data Availability Statement:** The data underlying this study have been uploaded to Dryad and are accessible using the following DOI: 10.5061/dryad.5hqbzkh2j.

## Abstract

Normal brain segmentation is available via FreeSurfer, Vbm, and Ibaspm software. However, these software packages cannot perform segmentation of the brain for patients with brain tumors. As we know, damage from extracerebral tumors to the brain occurs mainly by way of pushing and compressing while leaving the structure of the brain intact. Three-dimensional (3D) imaging, augmented reality (AR), and virtual reality (VR) technology have begun to be applied in clinical practice. The free medical open-source software 3D Slicer allows us to perform 3D simulations on a computer and requires little user interaction. Moreover, 3D Slicer can integrate with the third-party software mentioned above. The relationship between the tumor and surrounding brain tissue can be judged, but accurate brain segmentation cannot be performed using 3D Slicer. In this study, we combine 3D Slicer and FreeSurfer to provide a novel brain segmentation method for extracerebral tumors. This method can help surgeons identify the "real" relationship between the lesion and adjacent brain tissue before surgery and improve preoperative planning.

## Introduction

Extracerebral tumors are most typically benign, but some have malignant tumor growth characteristics, such as pushing, compressing, and even eroding adjacent normal brain tissue. Surgical treatment of these tumors is effective, but the prognosis depends largely on the location of the tumor and the degree of brain protection. The correct identification of brain functional areas during surgery helps surgeons to reduce the damage to normal brain tissue. Currently, 3D imaging technology can identify the relationship between extracerebral tumors and brain tissue by determining anatomical locations, which has provided great convenience for surgeons to make preoperative plans [1].

However, it is difficult to accurately identify brain areas, especially the motor function regions, because they are often deformed by the tumor. The solution is to use a neural navigation system, an anesthesia-arousal technique, and intraoperative electrophysiological monitoring during the operation. However, the above technologies are expensive and complex and thus cannot be performed in primary units. Here, we provide a cheap and simple alternative to localize brain functional areas covered by extracerebral tumors, which can be achieved in any units.

**Funding:** The authors of this study received no specific funding for this work. The hospital of Chengdu University of Traditional Chinese Medicine approved the use of their software for the experiments. The hospital had no role in study design, data collection and analysis, decision to publish, or preparation of the manuscript.

**Competing interests:** The authors declare that they have no conflicts of interest.

# Materials and methods

## Study design and study population

All patients were older than 5 years per the requirements of FreeSurfer [2]. The patients underwent magnetic resonance imaging (MRI) examination at the Hospital of Chengdu University of Traditional Chinese Medicine from June 2018 to July 2019 for diagnosing extracerebral intracranial tumors. Patients with severe peritumor edemas were excluded.

The Hospital of Chengdu University of Traditional Chinese Medicine Research Ethics Committee approved the study. All procedures involving human participants were in accordance with the ethical standards of the institutional and/or national research committee and with the 1964 Helsinki declaration and its later amendments or comparable ethical standards, and informed consent was not required because of the retrospective nature of this study.

## Data acquisition

Twenty-three patients with extracerebral tumors were selected, and they underwent nonenhanced T1-weighted (T1W) 3D fast spoiled gradient recalled (FSPGR) sequence MRI scans. The data were acquired on a 3.0 Tesla MRI scanner (GE Discovery MR750, America) using a 32-channel head or 8-channel standard joint head and neck coil. The scan parameters were as follows: TR/TE, 8.2 s/3.2 s; slice thickness, 0.5mm; FOV, 240 mm×240 mm; matrix, 256 mm×256 mm; flip angle, 12˚; 312 sections. The scans were performed mainly in the axial view, but some cases had scans performed in the coronal view.(URL: https://datadryad.org/stash, doi:10.5061/dryad.5hqbzkh2j) (temporary link: https://datadryad.org/stash/share/kjhTMQ7WsqzkOxnqsCAeAd0466sbUNU-0M1ElOfxpuM).

## Software

The 3D Slicer (http://www.slicer.org, Surgical Planning Laboratory, Harvard University, Boston, MA, USA, version 4.10.2) and FreeSurfer (http://www.freesurfer.net, MIT Health Sciences & Technology, and Massachusetts General Hospital, USA, version stable 6.0) software used in this study were free and open source. 3D-Slicer is available on multiple operating systems (Windows-64 bit, Linux, and MacOSX) and is used by surgeons for performing interactive segmentation, visualizing 3D medical images, and guiding therapy [3,4]. FreeSurfer requires Linux or MacOSX, either natively or a Windows-based virtual machine and is used mainly for the analysis and visualization of structural and functional neuroimaging data from patients with functional neurological disorders (FNDs) of the brain without structural damage, such as anxiety/depression, obsessive-compulsive disorder (OCD), epilepsy, posttraumatic brain syndrome (PTS) and so on [5]. For the hardware platform, we used a laptop with a 2.6 GHz Intel Core i7-9750 CPU, 16 GB RAM, and a 4GB NVIDIA GeForce GTX 1650 GPU running Windows 10 Home Edition 64-bit Version for 3D Slicer. Another desktop computer with a 3.2 GHz Intel Core i5-9750 CPU, 8 GB RAM, and an Integrated Graphics 620 GPU running Ubuntu 16.04 LTS 64-bit Version for FreeSurfer.

Dcm2nii is a part of the MRIcroN software (https://www.nitrc.org/projects/mricron) that can convert original DICOM data into the NIfTI or another medical image format, reducing the duration for importing data for operation [6].

## Fill between slices and GrowCut segmentation in 3D slicer

Fill between slices is a function of the Segment Editor module in 3D Slicer, which can perform a complete segmentation on selected slices using any editor effect. GrowCut Segmentation is an interactive segmentation approach that is based on the idea of a cellular automaton and is

another function of the Editor module. Egger et al. [7,8] indicate that GrowCut is a better way to perform tumor segmentation, as it is semiautomated, saves time, reduces operator effort and produces a more accurate segmentation than manually outlining the tumor slice by slice.

### Volume clip with model and swiss skull stripper in 3D slicer

Volume clip with model is a module of the VolumeClip extension, which needs to be installed in the Extensions Manager. This module can remove the volume contents inside or outside the selected surface model. Swiss Skull Stripper is another module that must be installed in Extensions Manager that can strip the scalp and skull from the brain, but a template (Atlas Image and Atlas Mask) is needed as a reference image and can be downloaded from the 3D Slicer website (https://www.slicer.org/wiki/Documentation/Nightly/Modules/SwissSkullStripper).

### Recon-all command in FreeSurfer

Recon-all (http://surfer.nmr.mgh.harvard.edu/fswiki/recon-all) is a standard and simple command in FreeSurfer that can automatically process whole-brain segmentation without human intervention step by step. Additionally, if a phase of the brain segmentation fails, the user can continue processing from that phase with the "Recon-all-X" command. The results produced by Recon-all include eight directories; in our study, we only needed the label, mri, and surf directories. The processing time depends on the computer configuration.

### Processing steps

The 3D Slicer software was used to remove the tumor manually, and brain segmentation was automatically performed according to the following steps in FreeSurfer (Fig 1):

- The 3D-T1W FSPGR DICOM data were dragged and dropped into the Dcm2nii software, converting the DICOM data to the SPM8 (3D NIFT) format. A new set of data, prefixed with the letters CO (remove neck), were generated and selected.

- The Nii data were loaded into 3D Slicer. Of particular note, different sequences can be overlaid only if they are in the same space. In this step, the "Centered" must be checked or the volume information section in the Volume module must be modified to ensure the standard spatial position.

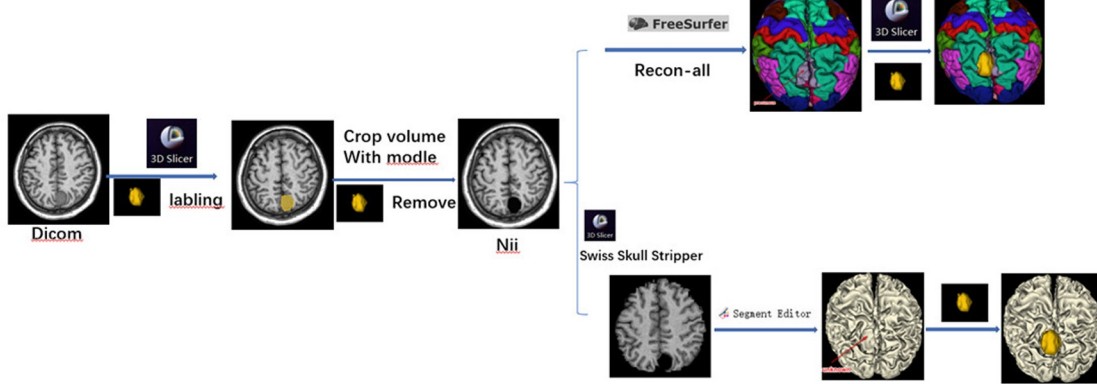

**Fig 1. Technology Roadmap.**

- An axial image was chosen and magnified to draw a boundary around the tumor. The Fill between slices or GrowCut Segmentation function was used to label the tumor and then show the tumor model in a 3D view. After inspection of the model, morphological operations from Segment Editor such as dilation, erosion, island removal, Scissors, and smoothing were used for postprocessing.

- The Segmentation module was used to export the tumor label to the model, and the Volume clip with model module was used to remove the tumor from the Nii data. Unchecking the "Clip outside" option and inputting a value of "0" (the typical background voxel brightness in MRI) in the pipeline box then generated a new set of Nii data without the tumor.;

- After loading the new Nii data into FreeSurfer, the "Recon-all" command was run to automate the segmentation of whole-brain structures; the duration of this operation depends on the machine configuration. Another traditional segmentation method uses the Swiss Skull Stripper module to strip the skull and scalp from the brain and then conduct a 3D reconstruction.

- According to the different needs of the brain templates, we can chose the different kind of files with the suffixes pial, anno and mgz. In our case, lh/rh.pial, lh/rh.aparc.anno and the original 3D-T1W Nii data files were selected and imported into 3D Slicer in sequence. Then, imported the aparc+aseg.mgz file and selected "Label Map" and "Centered" and the option to show "FreeSurferLabels".

- If a partial brain defect was found in the brain model, it was repaired in 3D Slicer with information from the unaffected side of the brain. Finally, the tumor label was reloaded and automatically superimposed onto the brain model to fuse the tumor and brain models.

## Model evaluation

Subjective evaluation was conducted according to clinical requirements. The cerebral cortex model and 3D volume-rendered images of the patients were both evaluated simultaneously by 2 senior and 2 younger neurosurgeons, and scored according to evaluation criteria (Table 1). If the scores were inconsistent, the final scores were determined through negotiation (Table 2). The two groups of scores were tested using two-tailed Wilcoxon signed-rank test. SPSS (version 17.0; SPSS Inc., Chicago, Illinois, USA) was used for analysis, and a p value<0.05 was considered statistically significant.

## Illustrative cases

**Sub6 and Sub8.**  A 69-year-old man who had decreased visual acuity and bitemporal hemianopia for 3 months and a 36-year-old man with 2-months progressive hearing loss were admitted to our department. They were diagnosed having pituitary tumor (Fig 2A)

**Table 1. Evaluation criteria ratings.**

| Scores | Quality | Interfere with the Scale |
|---|---|---|
| 3 | Good | The sulci and gyri can be clearly identified, and the functional area of the brain where the tumor is located can be identified |
| 2 | Medium | The sulci and gyri can be clearly identified, but the functional area of the brain where the tumor is located cannot be identified |
| 1 | Bad | The sulci and gyri cannot be clearly identified, and the functional area of the brain where the tumor is located cannot be identified |

**Table 2. Summary of the results.**

| Sub | Tumor Classification | Sex | Age | Tumor size (cm$^3$) | Brain Model Defect | Time (h) | Freesurfer Model Score (Senior/ Younger) | 3D Slicer Model Score (Senior/ Younger) |
|---|---|---|---|---|---|---|---|---|
| 1 | Pituitary adenoma | Female | 18 | 9.086 | No defect | 9.657 | 3/3 | 3/3 |
| 2 | Pituitary adenoma | Female | 41 | 11.835 | No defect | 9.865 | 3/3 | 3/3 |
| 3 | Pituitary adenoma | Female | 35 | 10.569 | No defect | 9.946 | 3/3 | 3/3 |
| 4 | Pituitary adenoma | Female | 30 | 8.357 | No defect | 9.609 | 3/3 | 3/3 |
| 5 | Pituitary adenoma | Female | 45 | 9.268 | No defect | 8.959 | 3/3 | 3/3 |
| 6 | Pituitary adenoma | Male | 69 | 8.542 | No defect | 9.257 | 3/3 | 3/3 |
| 7 | Pituitary adenoma | Male | 75 | 10.289 | No defect | 9.268 | 3/3 | 3/3 |
| 8 | Acoustic neuroma | Male | 36 | 9.039 | No defect | 9.454 | 3/3 | 3/2 |
| 9 | Acoustic neuroma | Male | 48 | 10.261 | No defect | 9.658 | 3/3 | 2/2 |
| 10 | Acoustic neuroma | Male | 70 | 16.078 | No defect | 8.579 | 3/3 | 2/2 |
| 11 | Acoustic neuroma | Female | 68 | 8.661 | No defect | 8.783 | 3/3 | 2/2 |
| 12 | Acoustic neuroma | Female | 54 | 8.198 | No defect | 9.236 | 3/3 | 2/2 |
| 13 | Trigeminal schwannoma | Male | 28 | 15.358 | No defect | 10.258 | 3/3 | 3/2 |
| 14 | Parafalcine meningioma | Male | 45 | 23.587 | Partial defect | 9.529 | 2/2 | 2/1 |
| 15 | Convexity meningioma | Female | 70 | 74.154 | Partial defect | 10.263 | 2/2 | 2/1 |
| 16 | Parafalcine meningioma | Female | 53 | 5.838 | No defect | 8.257 | 3/3 | 2/1 |
| 17 | Sella meningioma | Male | 53 | 8.404 | No defect | 8.314 | 3/3 | 2/2 |
| 18 | Sella meningioma | Female | 38 | 9.230 | No defect | 8.693 | 3/3 | 2/2 |
| 19 | Tentorial meningioma | Female | 56 | 3.673 | No defect | 8.563 | 3/3 | 1/1 |
| 20 | Sphenoid ridge meningioma | Male | 65 | 25.786 | Partial defect | 9.458 | 2/1 | 1/1 |
| 21 | Sphenoid ridge meningioma | Male | 49 | 19.258 | Partial defect | 9.568 | 2/1 | 1/1 |
| 22 | Craniopharyngioma | Female | 48 | 16.561 | No defect | 9.244 | 3/3 | 3/2 |
| 23 | Craniopharyngioma | Male | 44 | 15.854 | No defect | 10.059 | 3/3 | 3/2 |

and acoustic neuroma (Fig 3A) respectively according to the MRI-T1-3D-FSPGR examination. The tumors were labeled and removed by 3D Slicer (Figs 2B/2C and 3B/3C), and the brain segmentation was successfully operated by FreeSurfer (Figs 2D and 3D). The gyrus rectus is clearly observed compressed by the pituitary tumor (Fig 2E and 2F) and the junction of cerebellar hemisphere and brainstem are compressed by acoustic neuroma in the color brain model (Fig 3E and 3F). These models provided more intuitive information to the surgeons than conventional 3D VR images.

**Sub15.** A 70-year-old woman with giant convexity meningioma in the motor area had a limb movement disorder for 6 months (Fig 4A). The tumor was labeled and removed by 3D Slicer (Fig 4B and 4C). The brain tissue was segmented with the method we provide (Fig 4D). The gyrus around the tumor was segmented by FreeSurfer (Fig 4E) and repaired in 3D Slicer with reference to the opposite normal brain tissue (Fig 4F and 4G). The adjacent anterior central gyrus and posterior central gyrus are deformed by tumor extrusion, but they can still be identified after FreeSurfer segmentation (Fig 4F). After removing the tumor during the operation, the surgeon can identify the regions of the brain that have been deformed according to the color brain model (Fig 4H).

## Results

A total of 19 cases successfully underwent brain segmentation without errors. Partial gyri defects were found in 4 cases and repaired with information from the unaffected side of the

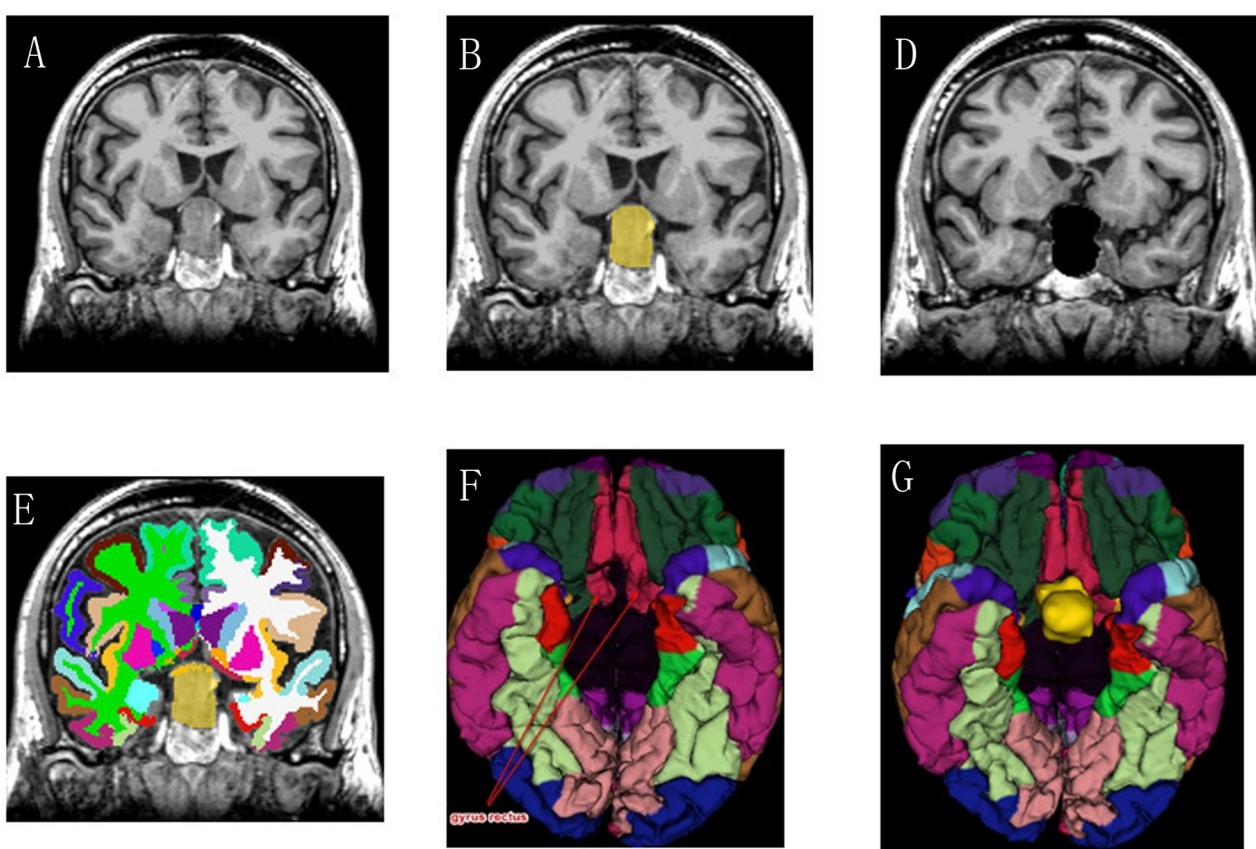

**Fig 2. (Sub 6, Pituitary adenoma).** These screenshots present the segmentation results in a coronal (A-C) slice for the automatic GrowCut tumor segmentation algorithm in 3D Slicer and automatic brain segmentation in FreeSurfer. The bilateral gyrus rectus was pushed and deformed by the tumor (D-F).

brain in 3D Slicer. The detailed results of our study are presented in Table 2. The total time consumed was approximately 9.325±0.586 hours. Both senior and younger neurosurgeons agreed that the brain model created by FreeSurfer was better and had more easily identifiable substructures (p<0.01).

## Discussion

Extracerebral intracranial tumors are usually benign [9] and have a clear interface with the surrounding brain tissue; complete removal of the tumor while minimizing neurological deficits should be pursued, especially for tumors in functional areas of the brain. 3D medical imaging processing software such as 3D Slicer, ITK-SNAP [10], and Mimics [11], can be used to locate the tumor and perform simulated surgery, but it is difficult to accurately distinguish the brain regions around the tumor only by the naked eye. At present, the solution to this problem is to use an intraoperative navigation system [12] combined with anesthesia-arousal techniques [13] to accurately identify the functional areas of the brain.

Neurosurgery relies on stereospecific neuronavigation devices for real-time and accurate positioning, which play a central role in modern neurosurgery. Although the methods mentioned above are accurate and effective, the equipment needed is complex and expensive, and doctors are required to master the complex operating procedures and maintain strict control

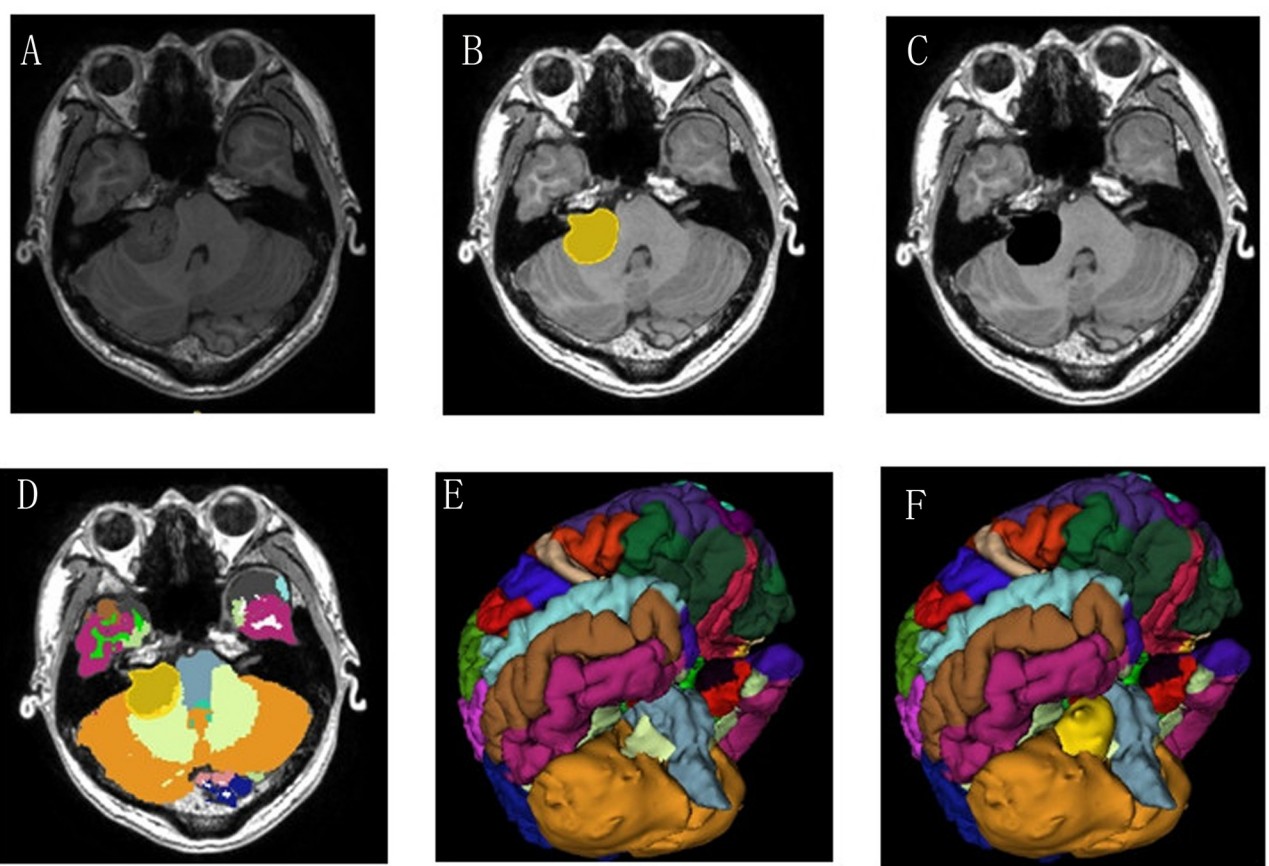

**Fig 3. (Sub 8, Right Acoustic neuroma).** These images depict the segmentation results in an axial (A-C) slice for manual tumor segmentation in 3D Slicer and automatic segmentation of brain structures without the tumor in FreeSurfer. The brainstem and cerebellum were pushed by the tumor (D-F).

of anesthesia depth. Hence, it is not a technology that can be carried out in hospitals at all levels, especially in developing countries [14]. Locating intracranial lesions accurately without this equipment is a difficult and urgent problem that needs to be solved.

The algorithms involved in brain tumor segmentation are varied and complex. Chang et al. [15] developed a deep learning algorithm to automatically segment gliomas, but this is a difficult technique for clinicians with no computer background and is therefore difficult to widely implement. A simpler and more practical segmentation method is needed. 3D Slicer is simple to operate, easy to learn and can accurately locate the lesion site, calculate the lesion volume, simulate the surgical path, and support virtual reality (VR) technology. It also has a variety of powerful tumor profiling tools [4]. Sina Application is a free medical image projection software that supports the Android system. The image can be projected onto the patient's head to accurately position intracranial lesions in real-time [16].

Combining 3D Slicer and the Sina software, Chen et al. [3] successfully performed the accurate localization of intracranial lesions and observed the relationship between the lesions and peripheral blood vessels. The above study indicates that even without stereotactic neuronavigation equipment, 3D Slicer combined with the Sina software can still provide neurosurgeons with a large amount of 3D information for the lesion and simulate reasonable surgical plans before surgery, which can greatly reduce surgical complications and improve the benefits to the patients. However, surgeons can still only locate intracranial lesions and determine their

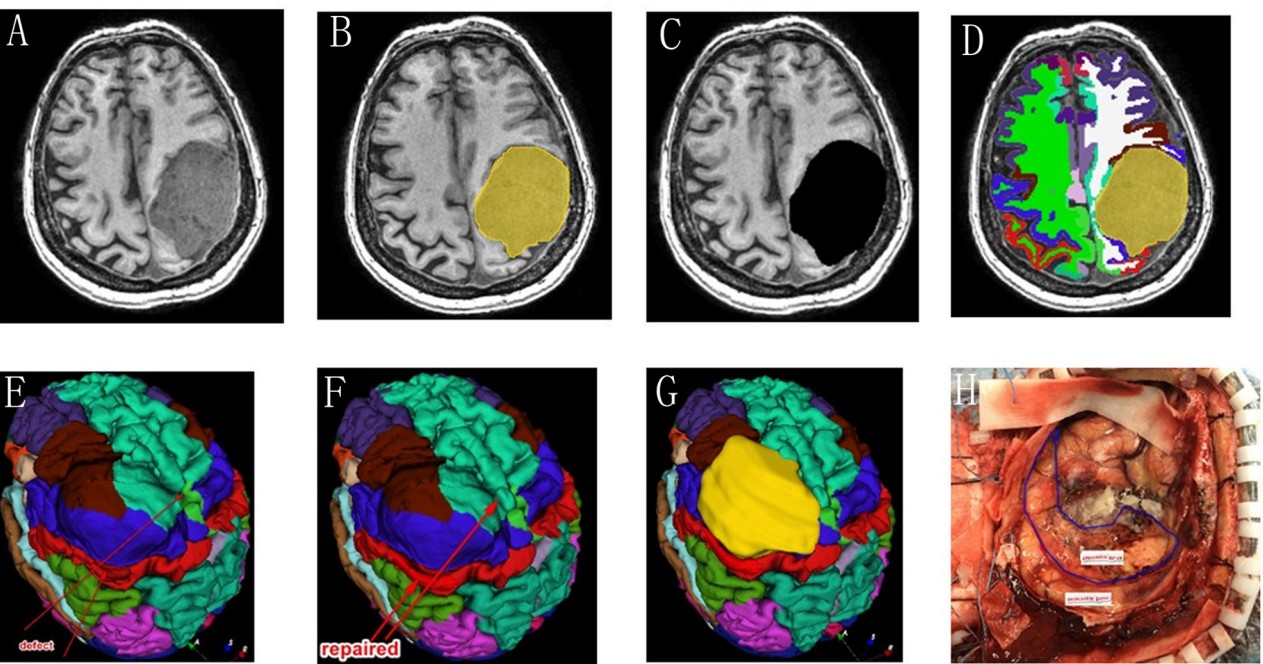

**Fig 4. (Sub 15, left convexity meningioma).** These images show the segmentation results in an axial slice for manual tumor segmentation in 3D Slicer (A-C) and automatic segmentation for the whole brain in FreeSurfer (D). The model was partially damaged (E), and the defect was repaired with information from the unaffected side of the brain (F). The postcentral, precentral, and superior frontal gyri and the caudal middle frontal cortex were pushed by the tumor (G). According to the above partitions, the "real" lobes can be identified after tumor removal (H).

relationship with adjacent tissues based on anatomical maps, and the results will inevitably have some errors.

FreeSurfer is another free program that performs segmentation of healthy brain structural and functional areas [2]. The generated data can be imported into 3D Slicer for localizing brain functional areas in the vicinity of intracranial tumors. However, according to the handbook, FreeSurfer is not suitable for segmenting a brain with an abnormal structure [2]. In order for FreeSurfer to construct cerebral cortex and deep white matter models, the complete gray-matter interface is required. For general neuroscience studies, this problem is irrelevant; however, the presence of intracranial lesions results in damage to the gray-matter interface. Therefore, for such diseases, FreeSurfer cannot directly segment the brain model, making the software results unsuitable for clinical neurosurgeons. Therefore, up to now, this software has only been used in the field of functional neurological diseases, such as hemifacial spasm (HFS) [17], epilepsy [18] and posttraumatic head syndrome (PTSD) [19].

The algorithm of FreeSurfer is very complex; briefly, its processing includes motion correction and averaging [20] of multiple volumetric T1-weighted images (when more than one is available), removal of nonbrain tissue using a hybrid watershed/surface deformation procedure [21], automated Talairach transformation, and segmentation of the subcortical white matter and deep gray matter volumetric structures. In our study, we proposed a new, simple method to remove tumors from the T1W 3D-FSPGR data of patients with extracerebral tumors via a virtual operation by setting the values corresponding to the tumor to "0". The interface between the normal brain tissue and the tumor is artificially divided. Then, the new data are imported into FreeSurfer to automatically segment the brain structure. FreeSurfer can automatically identify the "0" regions as nonbrain tissue to be excluded.

The result of this method is feasible, and although some brain tissue defects may occur (mostly in cases involving supratentorial extracerebral tumors), 3D Slicer can be used to repair them. Sub 15 is a typical case with a giant convexity meningioma in the motor area (Fig 4). After removing the tumor during the operation, the surgeon can identify the regions of the brain that have been deformed according to the color brain model. This method does not require mastery of complex computer languages and operating procedures, allowing even regular residents to operate the software.

Based on the procedures from Chen et al. [3], our study innovatively utilizes 3DSlicer and FreeSurfer software to break through the conventional limitations of FreeSurfer; our technique can not only realize the accurate positioning of intracranial lesions and brain functional areas in real-time but also assist physicians in assessing the relationship between the tumor and peripheral brain regions. Both the younger and senior physicians agreed that the color brain model created from our method was more easily recognizable and understood than the traditional single-tone brain model. In addition, the above software is free, can be run flexibly at all levels of units and is not restricted by expensive medical equipment. Moreover, the STL (stereolithography) file format created by 3D Slicer can be used to 3D print brain models, aiding AR/VR virtual surgery and VR surgical teaching and providing more accurate 3D information for increasing the confidence of surgeons, allowing them to make better surgical plans and better train young surgeons, as well as shortening their learning curves [1].

There are several future works in development. We plan to conduct further multimodal image fusion studies based on the current research; for example, the integration of diffusion tensor imaging (DTI) into the 3D brain model created by FreeSurfer could lead to a more accurate judgment of the relationship between the tumor and surrounding nerve fiber bundles and help surgeons better protect white matter tracts (WMT) [22,23]. In addition, vascular (CTA/MRA, MRV) and skull (CT) models can be fused with the FreeSurfer models, and combined with Sina, these models could help surgeons make more accurate preoperative plans and yield more benefit for surgeon training and patient education. 3D printing is a further derivation of VR technology, and color 3D printing is more suitable for illustrating functional areas of the brain, while its costs need to be reduced and its efficiency needs to be improved [24].

In summary, the methods we have provided can be used by surgeons as a cheap and easy way to identify the "real" relationship between the extracerebral intracranial lesion and adjacent brain tissue before surgery and improve preoperative planning in any unit.

## Conclusions

Our approach for segmenting extracerebral intracranial tumors is facilitated, requires no complex programming, and can be utilized even by doctors without a computer background independently. The resulting clear 3D color structure image provides additional assistance to surgeons for obtaining more accurate information and making better preoperative planning. Neurosurgery is a subject closely integrated with computer and artificial intelligence. Proficiency in computer-related medical technology is an inevitable skill for neurosurgeons to develop [25].

## Limitations

Our subjects were limited to those with extracerebral tumors with no significant edema, brain tissue deformations only by tumor compression, and a lack of brain defects. Cases involving tumors with significant edema or in the intracerebral region tend to present with destruction of the white or gray matter structure, and thus this segmentation method is no longer applicable. Moreover, our case involves a small sample size, and future studies will require a larger

group of patients. Additionally, data processing in FreeSurfer is time consuming, inefficient and may even produce errors that require human intervention. Our method can only provide anatomical segmentation of brain tissue according to different brain structure templates.

## Supporting information

**S1 File.**
(ZIP)

**S2 File.**
(ZIP)

**S3 File.**
(ZIP)

**S4 File.**
(ZIP)

**S5 File.**
(ZIP)

**S6 File.**
(ZIP)

## Author Contributions

**Conceptualization:** Dongdong Yang.

**Data curation:** Xiaolin Hou, Meijun Liu, Yuan Zhou, Min Shi.

**Investigation:** Dingjun Li.

**Software:** Xiaolin Hou.

**Supervision:** Xiaolin Hou, Yuan Zhou.

**Writing – original draft:** Xiaolin Hou.

**Writing – review & editing:** Dongdong Yang.

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
