## [Decision Letter · Decision Letter 0]

25 Nov 2019

PONE-D-19-26376

A New Simple Brain Segmentation Method for Extracerebral Intracranial Tumors

PLOS ONE

Dear Mrs dongdong,

Thank you for submitting your manuscript to PLOS ONE. After careful consideration, we feel that it has merit but does not fully meet PLOS ONE’s publication criteria as it currently stands. Therefore, we invite you to submit a revised version of the manuscript that addresses the points raised during the review process.

We would appreciate receiving your revised manuscript by Jan 09 2020 11:59PM. To enhance the reproducibility of your results, we recommend that if applicable you deposit your laboratory protocols in protocols.io, where a protocol can be assigned its own identifier (DOI) such that it can be cited independently in the future. For instructions see: http://journals.plos.org/plosone/s/submission-guidelines#loc-laboratory-protocols

We look forward to receiving your revised manuscript.

Kind regards,

Jonathan H Sherman

Academic Editor

PLOS ONE

Journal Requirements:

2. Thank you for including your ethics statement: The study was approved by the medical ethical committee of our hospital, and informed consent was deserted because of the retrospective collection of this study.

No

a) Please provide an amended Funding Statement that declares *all* the funding or sources of support received during this specific study (whether external or internal to your organization) as detailed online in our guide for authors at http://journals.plos.org/plosone/s/submit-now.  

b) Please state what role the funders took in the study.  If any authors received a salary from any of your funders, please state which authors and which funder. If the funders had no role, please state: "The funders had no role in study design, data collection and analysis, decision to publish, or preparation of the manuscript."

5. Please ensure that you refer to Figures 2 and 3 in your text as, if accepted, production will need this reference to link the reader to the figure.

Reviewers' comments:

Reviewer's Responses to Questions

**Comments to the Author**

1. Is the manuscript technically sound, and do the data support the conclusions?

Reviewer #1: Yes

Reviewer #2: Partly

2. Has the statistical analysis been performed appropriately and rigorously? 

Reviewer #1: I Don't Know

Reviewer #2: N/A

3. Have the authors made all data underlying the findings in their manuscript fully available?

Reviewer #1: No

Reviewer #2: No

4. Is the manuscript presented in an intelligible fashion and written in standard English?

Reviewer #1: No

Reviewer #2: No

5. Review Comments to the Author

Reviewer #1: There is a lot of statistics hidden in the software that is not addressed in the methods. To my impression, these are pre-processed algorithms provided by the manufacturers, which are not clearly presented in the methods section, other than that they are being applied.

The manuscript in my opinion serves best the purpose of a methods paper. However, there is no other validation of the proposed method of inquiry other than the visual confirmation of realistic segmentation by the algorithm based on clinicians estimation.

Throughout the paper there needs to be attention for the use of grammar and typos.

The real valor of these segmentation algorithms would be in the application of pre-surgical planning for tumors with a lot of edema of intraparenchymal invasion. The manuscript would benefit with regard to scientific and translational impact if the authors could demonstrate any feasibility of these algorithms or the applied software to tackle these hurdles.

Reviewer #2: This manuscript is a methodology paper explaining the methods for reconstruction of MRI data for anatomic neurosurgical planning using free, open-source software. This methodology is useful for the field of neurosurgery, and is both less technically challenging and less financially restrictive than other options. This method may potentially be a useful addition to the neurosurgical toolbox, but I find a number of problems with this submission. Primarily, the data used to generate their figures is not made available, and as such their figures cannot be reproduced to validate their methodology. Secondarily, this article could benefit from a significant overhaul by a professional, English-language scientific editor. There are sections of the manuscript where the meaning of the authors is unclear, there are multiple incomplete sentences, and innumerable other grammatical and syntactical errors.

6. PLOS authors have the option to publish the peer review history of their article (what does this mean?). If published, this will include your full peer review and any attached files.

Reviewer #1: Yes: Rutger Balvers

Reviewer #2: No

---

## [Author Response · Author response to Decision Letter 0]

14 Jan 2020

Dear Editor,

We have studied the valuable comments from you, the assistant editor and reviewers carefully, and tried our best to revise the manuscript. We want to upload all the patient data, but because the Dicom/Nii data of all the patients is too large, we can only upload part of the patient data to verify the repeatability of our study. The point to point responds to the reviewer’s comments are listed as following: 

Responds to the reviewer’s comments:

Reviewer 1 

Comment 1: There is a lot of statistics hidden in the software that is not addressed in the methods. To my impression, these are pre-processed algorithms provided by the manufacturers, which are not clearly presented in the methods section, other than that they are being applied.

Response: According to the reviewer’s comment, we have added the software instructions, the algorithm of FreeSurfer is very complex; briefly, its processing includes motion correction and averaging of multiple volumetric T1-weighted images (when more than one is available), removal of nonbrain tissue using a hybrid watershed/surface deformation procedure, automated Talairach transformation, and segmentation of the subcortical white matter and deep gray matter volumetric structures. As a clinician, it is difficult to master the running algorithm of the software, and the official website does not mention too much. We were inspired by the simple calculation process provided by the official website. For the extracerebral intracranial tumors without serious peripheral edema, the boundary was clearly separated from the normal brain tissue. After the tumor was artificially labeled and removed, the interface between the normal brain tissue and the tumor is artificially divided. Then, the software could successfully carry out automatic brain tissue segmentation according to its original algorithm.

Comment 2: The manuscript in my opinion serves best the purpose of a methods paper. However, there is no other validation of the proposed method of inquiry other than the visual confirmation of realistic segmentation by the algorithm based on clinicians estimation.

Response: Thank you for your suggestion. We divided the definition of the sulci and gyri into three levels, the cerebral cortex model created by FreeSurfer and 3D volume-rendered images created by 3D Slicer were both evaluated simultaneously by 2 senior and 2 younger neurosurgeons, and scored according to subjective evaluation criteria. The two groups of scores were tested using a two-tailed Wilcoxon signed-rank test. SPSS was used for analysis, both senior and younger neurosurgeons agreed that the brain model created by FreeSurfer was better and had more easily identifiable substructures (p<0.01).

Comment 3: Throughout the paper there needs to be attention for the use of grammar and typos.

Response: Thank you for your careful work. We have submitted the original manuscript to the English translation company for revision.

Comment 4: The real valor of these segmentation algorithms would be in the application of pre-surgical planning for tumors with a lot of edema of intraparenchymal invasion. The manuscript would benefit with regard to scientific and translational impact if the authors could demonstrate any feasibility of these algorithms or the applied software to tackle these hurdles.

Response: Thank you for your advice. However, at present, there is no algorithm that can automatically segment the normal brain tissue from patients with brain tumors. Not just intracranial tumors cannot be segmented, but also extracranial tumors with severe peripheral edema also could not be segmented, because the blood brain barrier has been broken and gray matter boundary is not clear, the task of brain segmentation cannot be completed by Freesurfer software. We have tried to use Freesurfer software to segment the extracranial tumors with severe edema and intracranial tumors, unfortunately, it can only get a defect model of brain tissue or even produce software errors that prevent brain segmentation. Therefore, the proposed method is currently only applicable to brain segmentation of extracranial tumors. Nevertheless, this method can still provide useful information for clinicians, especially when the extracranial tumor is located in a motor and language function areas.

Reviewer 2

Comment 1: Primarily, the data used to generate their figures is not made available, and as such their figures cannot be reproduced to validate their methodology. 

Response: According to the reviewer’s comment, we want to upload all the patient data, but because the Dicom/Nii data of all the patients is too large, we can only upload part of the patient data to verify the repeatability of our study.

Comment 2: Secondarily, this article could benefit from a significant overhaul by a professional, English-language scientific editor. There are sections of the manuscript where the meaning of the authors is unclear, there are multiple incomplete sentences, and innumerable other grammatical and syntactical errors.

Response: Thank you for your careful work. We have submitted the original manuscript to the English translation company for revision.

---

## [Decision Letter · Decision Letter 1]

3 Feb 2020

PONE-D-19-26376R1

A New Simple Brain Segmentation Method for Extracerebral Intracranial Tumors

PLOS ONE

Dear Mrs dongdong,

Thank you for submitting your manuscript to PLOS ONE. After careful consideration, we feel that it has merit but does not fully meet PLOS ONE’s publication criteria as it currently stands. Therefore, we invite you to submit a revised version of the manuscript that addresses the points raised during the review process.

We would appreciate receiving your revised manuscript by Mar 19 2020 11:59PM. To enhance the reproducibility of your results, we recommend that if applicable you deposit your laboratory protocols in protocols.io, where a protocol can be assigned its own identifier (DOI) such that it can be cited independently in the future. For instructions see: http://journals.plos.org/plosone/s/submission-guidelines#loc-laboratory-protocols

We look forward to receiving your revised manuscript.

Kind regards,

Jonathan H Sherman

Academic Editor

PLOS ONE

Reviewers' comments:

Reviewer's Responses to Questions

**Comments to the Author**

1. If the authors have adequately addressed your comments raised in a previous round of review and you feel that this manuscript is now acceptable for publication, you may indicate that here to bypass the “Comments to the Author” section, enter your conflict of interest statement in the “Confidential to Editor” section, and submit your "Accept" recommendation.

Reviewer #2: All comments have been addressed

Reviewer #3: All comments have been addressed

2. Is the manuscript technically sound, and do the data support the conclusions?

Reviewer #2: Partly

Reviewer #3: Partly

3. Has the statistical analysis been performed appropriately and rigorously? 

Reviewer #2: N/A

Reviewer #3: I Don't Know

4. Have the authors made all data underlying the findings in their manuscript fully available?

Reviewer #2: Yes

Reviewer #3: No

5. Is the manuscript presented in an intelligible fashion and written in standard English?

Reviewer #2: Yes

Reviewer #3: No

6. Review Comments to the Author

Reviewer #2: (No Response)

Reviewer #3: The manuscript would benefit from revision by an English language editor. The authors have made a portion of their data available. The segmentation of the lesions appear accurate but there is no way to validate without the primary imaging data, etc. There is no validation of the method internally by the authors other than a relative neurosurgeon rating score, which is not a validated measure. The argument that this technique can help identify function is only partially accurate. Neurophysiologcial monitoring and awake technique remain pillars of clinical care because anatomy does not always equal function. Receptive speech centers are a classic example. We know that clinically they are not restricted to angular gyrus and supramarginal gyrus but can be represented broadly in the inferior parietal lobule and posterior temporal lobe in the dominant hemisphere.

7. PLOS authors have the option to publish the peer review history of their article (what does this mean?). If published, this will include your full peer review and any attached files.

Reviewer #2: No

Reviewer #3: No

---

## [Author Response · Author response to Decision Letter 1]

16 Feb 2020

Dear Editor,

We have studied the valuable comments from you, the assistant editor and reviewers carefully, and tried our best to revise the manuscript. We have uploaded part of the patient data to verify the repeatability of our study.

The point to point responds to the reviewer’s comments are listed as following: 

Responds to the reviewer’s comments:

Reviewer 3

Comment 1: The manuscript would benefit from revision by an English language editor.

Response: Thank you for your careful work. We have submitted the original manuscript to the English translation company for revision.

Comment 2: The authors have made a portion of their data available. The segmentation of the lesions appear accurate but there is no way to validate without the primary imaging data, etc. 

Response: Thank you for your careful work. DICOM data is raw data, it contains the patient's privacy information. Therefore, we use dcm2nii software to transform DICOM data into “T1.nii” data, and remove the patient's privacy information. With the help of 3D Slicer，tumor was labeled and removed，the new data of “T1 without tumor. nii” is generated and can be loaded into Freesurfer software to segment the brain. 

Comment 3: There is no validation of the method internally by the authors other than a relative neurosurgeon rating score, which is not a validated measure. 

Response: Thank you for your advice. However, we are just trying to compare the difference of brain model that's been segmented by FreeSurfer software and traditional methods. The neurosurgeon rating score is the most direct and simple way to judge the pros and cons of the two models. In fact, the advantages of the former are obvious. Through literature review, there are similar evaluation methods, just like Karibe et al, in the early stage, used DWI to judge the damage degree of cerebral hemorrhage to corticospinal tract (CST), they also performed CST injury classification by visually observing the relationship between hematoma and CST. (Karibe H, Shimizu H, Tominaga T, et al. Diffusion-weighted magnetic resonance imaging in the early evaluation on of corticospinal tract injury to predict functional motor outcome in patients with deep intracerebral hemorrhage. J Neurosurg.2000;92(1):58-63.) Pandrangi et al. used patients' subjective scores to prove that VR technology is more advantageous in communicating with patients. (Pandrangi VC, Gaston B, Appelbaum NP, Albuquerque FC, Jr., Levy MM, Larson RA. The application of virtual reality in patient education. Ann Vasc Surg. 2019;59: 184-189.)

Comment 4: The argument that this technique can help identify function is only partially accurate. Neurophysiologcial monitoring and awake technique remain pillars of clinical care because anatomy does not always equal function. Receptive speech centers are a classic example. We know that clinically they are not restricted to angular gyrus and supramarginal gyrus but can be represented broadly in the inferior parietal lobule and posterior temporal lobe in the dominant hemisphere.

Response: Thank you for your advice. This is a mistake in writing. We have changed

the summary that the methods we have provided can be used by surgeons as a cheap and easy way to identify the “real” relationship between the extracerebral intracranial lesion and adjacent brain tissue before surgery and improve preoperative planning in any unit. Our method can only provide anatomical segmentation of brain tissue according to different brain structure templates.

---

## [Decision Letter · Decision Letter 2]

9 Mar 2020

A New Simple Brain Segmentation Method for Extracerebral Intracranial Tumors

PONE-D-19-26376R2

Dear Dr. dongdong,

We are pleased to inform you that your manuscript has been judged scientifically suitable for publication and will be formally accepted for publication once it complies with all outstanding technical requirements.

With kind regards,

Jonathan H Sherman

Academic Editor

PLOS ONE

Additional Editor Comments (optional):

Reviewers' comments:

Reviewer's Responses to Questions

**Comments to the Author**

1. If the authors have adequately addressed your comments raised in a previous round of review and you feel that this manuscript is now acceptable for publication, you may indicate that here to bypass the “Comments to the Author” section, enter your conflict of interest statement in the “Confidential to Editor” section, and submit your "Accept" recommendation.

Reviewer #3: All comments have been addressed

2. Is the manuscript technically sound, and do the data support the conclusions?

Reviewer #3: Yes

3. Has the statistical analysis been performed appropriately and rigorously? 

Reviewer #3: Yes

4. Have the authors made all data underlying the findings in their manuscript fully available?

Reviewer #3: No

5. Is the manuscript presented in an intelligible fashion and written in standard English?

Reviewer #3: Yes

6. Review Comments to the Author

Reviewer #3: The article is written in more clearly. They have addressed most of the comments. They apparently cannot provide deidentified imaging data for all patients.

7. PLOS authors have the option to publish the peer review history of their article (what does this mean?). If published, this will include your full peer review and any attached files.

Reviewer #3: No

---

## [Editor Report · Acceptance letter]

7 Apr 2020

PONE-D-19-26376R2 

A New Simple Brain Segmentation Method for Extracerebral Intracranial Tumors 

Dear Dr. Yang:

I am pleased to inform you that your manuscript has been deemed suitable for publication in PLOS ONE. Congratulations! Your manuscript is now with our production department. 

With kind regards,

on behalf of

Dr. Jonathan H Sherman 

Academic Editor

PLOS ONE